# Metastases and Recurrence Risk Factors in Endometrial Cancer—The Role of Selected Molecular Changes, Hormonal Factors, Diagnostic Methods and Surgery Procedures

**DOI:** 10.3390/cancers16010179

**Published:** 2023-12-29

**Authors:** Anna Markowska, Włodzimierz Baranowski, Kazimierz Pityński, Anita Chudecka-Głaz, Janina Markowska, Włodzimierz Sawicki

**Affiliations:** 1Department of Perinatology and Women’s Diseases, Poznan University of Medical Sciences, 60-535 Poznan, Poland; annamarkowska@vp.pl; 2Department of Gynecological Oncology, Military Institute of Medicine, 04-141 Warsaw, Poland; 3Department of Gynecology and Oncology, Jagiellonian University Medical College, 31-501 Krakow, Poland; pitynski@wp.pl; 4Department of Gynecological Surgery and Gynecological Oncology of Adults and Adolescents, Pomeranian Medical University, 70-204 Szczecin, Poland; anitagl@poczta.onet.pl; 5Gynecological Oncology Center Poznań, Poznanska 58A, 60-850 Poznan, Poland; jmarkmed@poczta.onet.pl; 6Department of Obstetrics, Gynecology and Gynecological Oncology, Medical University of Warsaw, 02-091 Warsaw, Poland; saw55@wp.pl

**Keywords:** endometrial cancer metastasis, TCGA classification, hormones, ultrasonography, hysteroscopy

## Abstract

**Simple Summary:**

This paper delves into the issue of metastatic endometrial cancer (EC), which significantly impacts treatment success and overall survival rates. The study explores various factors contributing to metastasis, including the molecular profile of EC, hormone activity (like estrogen and prolactin), and pro-inflammatory adipocytokines. It also investigates how altered microRNA expression affects gene regulation linked to the dissemination of the cancer. The paper highlights the importance of imaging techniques, particularly transvaginal ultrasound with tumor-free distance (uTFD), in detecting metastases. Additionally, it discusses how diagnostic and therapeutic methods can influence the spread of EC, with hysteroscopy potentially increasing risk in advanced stages and laparoscopic hysterectomy being safer if performed with care. This research could improve our understanding of EC and guide better diagnostic and treatment strategies.

**Abstract:**

The presence of metastatic endometrial cancer (EC) is a key problem in treatment failure associated with reduced overall survival rates. The most common metastatic location is the pelvic lymph nodes, and the least common is the brain. The presence of metastasis depends on many factors, including the molecular profile of cancer (according to the TCGA—Genome Atlas), the activity of certain hormones (estrogen, prolactin), and pro-inflammatory adipocytokines. Additionally, an altered expression of microRNAs affecting the regulation of numerous genes is also related to the spread of cancer. This paper also discusses the value of imaging methods in detecting metastases; the primary role is attributed to the standard transvaginal USG with the tumor-free distance (uTFD) option. The influence of diagnostic and therapeutic methods on EC spread is also described. Hysteroscopy, according to the analysis discussed above, may increase the risk of metastases through a fluid medium, mainly performed in advanced stages of EC. According to another analysis, laparoscopic hysterectomy performed with particular attention to avoiding risky procedures (trocar flushing, tissue traumatization, preserving a margin of normal tissue) was not found to increase the risk of EC dissemination.

## 1. Introduction

After cervical cancer, endometrial cancer (EC) is the second most common gynecological malignancy in women worldwide. According to the latest epidemiological data available worldwide in 2020, there were 417,367 new cases of cancer (accounting for 4.5% of all malignancies in women) and 97,370 deaths from this cancer. Moreover, both EC incidence and mortality rates have risen over the past decades. This problem is increasing, mainly in countries with high economic standards, where there has been an escalation in both obesity and the prevalence of type 2 diabetes; these factors have been well documented in relation to the development of EC [1,2,3,4]. Fortunately, most patients have localized disease when diagnosed; about 20% are diagnosed with regional spread, and 9% have distant metastases. According to the Surveillance, Epidemiology, and End Results (SEERs) database and other studies, a 5-year survival depends mainly on cancer spread. In the early stages, the five-year survival rate is more than 80%, then decreases to 56–69% in cases of locally ongoing spread, dropping to 17–20% in patients with distant metastasis [5,6]. The most common site of EC metastases is the pelvic lymph nodes. Their presence is crucial in predicting cancer recurrence [7,8,9]. Lymphatic mapping of the lymph nodes, including the sentinel lymph node technique, increases the rate of metastasis detection, but it should be stressed that more than 5% of results may be false negatives [10]. A retrospective study by Restaino et al. [11] involving 1576 patients with EC who underwent uterine resection with adnexa after lymph node mapping showed that the most common metastatic lesions were bilateral external iliac and obturator lymph nodes. The detection rate in this analysis was 93.4%. In a study by Byrak et al. [12] in 435 patients with EC, the rate of micrometastasis to the omentum with complete surgical staging was analyzed. They found that omental micrometastases correlated with positive peritoneal cytology and adnexa involvement in 5.3% of these patients. The rarest metastatic location is the brain; the percentage of metastases accounts for 0.48% of all metastatic EC lesions [13]. The subdivisions of EC according to the molecular classification using the Genome Atlas and imaging results increase the prognostic value.

The literature search was conducted systematically to identify relevant studies related to EC metastasis and recurrence risk factors. The analysis was conducted from May 2023 to July 2023 and included published literature from 1998 to 2023. Research on Pubmed, Web of Science, and Scopus was carried out. The search utilized a set of specific keywords and keyword combinations, including “endometrial cancer metastasis” and/or “TCGA classification” and/or “hormones” and/or “ultrasonography” and/or “hysteroscopy” and/or “laparoscopy”. Studies included in this review were carefully selected by all authors based on predefined inclusion and exclusion criteria. Inclusion criteria encompassed studies focusing on EC metastasis, molecular profiling according to TCGA, hormonal influences, adipocytokines, microRNA expression, hysteroscopy, and laparoscopy. Following the initial screening, the authors assessed complete versions of the chosen papers, extracting pertinent information related to the study’s characteristics and outcomes independently. A thorough examination of all references was conducted to identify any additional qualifying studies. The research encompassed retrospective case series, prospective trials, and randomized controlled trials (RCTs). Studies deemed incongruent with the study’s objectives, case reports, redundant investigations, incomplete manuscripts, and articles not in English were excluded from consideration.

## 2. Molecular Classification of Endometrial Cancer

The Cancer Genome Atlas was the first project to evaluate a large number of cancers using whole genome sequencing, including endometrial cancer [14]. It identified four subgroups of endometrial cancer with different molecular profiles: tumors with mutated DNA polymerase ε (POLE), tumors exhibiting microsatellite instability (MSI), and tumors with copy-number low (CNL) and copy number high (CNH). It was shown that molecular subgroups of EC patients divided according to Cancer Genome Atlas rules are closely related to clinical outcomes with these molecular subgroups. The original Genome Atlas used total genome sequencing for the evaluation of molecular changes, which is not clinically or economically feasible on a large population scale. However, much data confirmed that molecular analysis profiles using more feasible immunohistochemistry and more feasible sequencing techniques, such as NGS (new generation sequencing), could also be useful [15,16,17,18].

In particular, the ProMisE (Proactive Risk Classifier of Endometrial Cancer) technique has proven reliable for identifying tumor prognosis in the four subgroups of The Cancer Genome Atlas. 

The prognostic value of molecular subgroups, including additional molecular alterations, was assessed in the PORTEC 3 study using univariate and multivariate analysis with clinicopathological factors (age, grade of malignancy, lymphovascular space invasion (LVSI), and treatment, both in the general population and, in particular, analysis restricted to cases with high- and intermediate-risk groups. In both analyses, p53 mutation and the presence of LVSI were the most important prognostic factors for local recurrence, distant recurrence, and overall survival, while L1CAM (transmembrane protein member of the L1 protein family) expression in more than 10% of patients had prognostic value for distant recurrence and overall survival. After excluding cases with the favorable factors (POLE mutation) and the unfavorable factors listed above, the final analysis showed that MSI had prognostic value for distant recurrence and overall survival, and mutation status in exon 3 of the CTNNB1 genes had prognostic value for distant recurrence only [19].

Ashley et al. [20] showed that the genome of endometrial cancers with POLE and MSI molecular subtypes are shaped by specific mutational processes (such as loss of proofreading ability by polymerase and MSI), while endometrial cancers with low/high copy number subtypes and uterine sarcomas are dominated by mutational processes associated with aging. The analysis of mutational signatures carried out in this paper extends the molecular classification of TCGA, showing variations in mutational processes even among cancers of the same molecular subtype. In fact, it was found that 15% of MSI-overexpressing tumors do not exhibit a predominant MSI signature, while only 15% of copy-overexpressing tumors (with similar activity when compared to serous carcinomas) have a signature associated with a damaged DNA-repair system. In addition, it was observed that while the molecular subtype of EC according to the TCGA classification is generally stable between the primary tumor and metastasis, changes in the mutational signature occur in more than 25% of endometrial cancers, suggesting that additional defects in DNA repair mechanisms may influence the development of the tumor. Thus, mutational signatures and a thorough understanding of the mutagenic processes/pathways present in a given cancer may influence therapeutic options, especially in the absence of mutations or changes in the copy number of target genes (e.g., genomic features associated with DNA repair damage), even in the absence of BRCA1/BRCA2 inactivation. Significantly, given the observed changes in mutational signatures between the primary tumor and metastasis in a subset of cases, it is possible that in advanced disease, a genome-wide analysis should be performed in the metastatic tissue rather than the primary tumor.

## 3. Hormonal and Molecular Aspects

In the process of metastasis of any cancer, including endometrial cancer, three elements are important: disturbances in the adhesion of cells to the ECM (extracellular matrix) causing easy detachment from the substrate, disturbances in the cytoskeleton causing increased cell migration; and invasive properties of the cancer cell, which make implantation in a site other than the primary site (lymph nodes, parenchymal organs) possible [21,22,23].

Estrogens, through estrogen receptors (ER) and extragenomic interactions, cause the remodeling of actin, the main protein of the cytoskeleton, as well as the remodeling of the cell membrane. At the molecular level, this phenomenon depends on inducing phosphorylation on Thr(558) in the actin-binding protein moesin. This interaction causes increased migration and implantation capabilities of endometrial cells. Such effects are observed in studies of endometrial cell cultures (Ishikawa cells) and also in native endometrial cells (endometrial stromal cells—ESC). Similar effects to estradiol are caused by tamoxifen but not raloxifene. Clinically, this is manifested by endometrial stimulation and hyperplasia in women taking tamoxifen—women using raloxifene do not show such effects because selective estrogen receptor modulators (SERM) do not interact with moesin, so it does not change the cytoskeleton of endometrial cells. The molecular basis of this phenomenon is related to extragenomic modulation, in which estradiol and tamoxifen are involved. Furthermore, the activity of G proteins and Rho group-related kinases is modulated, directly affecting the migratory and proliferative properties of cells [23,24]. 

In other in vitro and in vivo studies, estradiol has been shown to promote endometrial cancer proliferation, migration, and invasion by activating the IL-6 pathway, which is involved in multiple signaling pathways associated with ER, Bcl-2, Cyclin D1, and MMP2 invasion [25].

Recently, the role of prolactin in the metastasis of endometrial cancer and its association with reduced sensitivity to chemotherapy has drawn particular attention. The mechanism of action of prolactin in stimulating the processes of cancer metastasis is quite complex and includes endocrine, paracrine, and autocrine mechanisms, including interactions with immune cells.

Two modes of prolactin action are considered: receptor and non-receptor. The anti-apoptotic effect of prolactin and PRLR is due to the blocking of the expression of Stat5a/b, which in turn results in increased expression of Bcl-2 protein (anti-apoptotic protein) and decreased expression of Bax protein (pro-apoptotic protein) and also by increasing the expression of the chaperone protein Hsp90A, which protects cells from apoptosis. Evidence for the proliferation-stimulating effect of prolactin is its synthesis in vascular endothelial cells and stromal cells. Additional evidence for the possible carcinogenic effect of prolactin is the possibility that its receptor (PRLR) interacts with various agonistic ligands. This creates a specific microenvironment in metastatic foci that promotes proliferative processes. In addition, prolactin has a direct stimulatory effect on the proliferation of vascular endothelial cells and also exerts an indirect effect by increasing the expression of VEGF and other proangiogenic factors [26,27]. 

The pro-inflammatory adipocytokines leptin, visfatin, and resistin are also involved in the progression and spread of endometrial cancer cells [22]. 

Studies on molecular mechanisms of metastasis in ECs have led to the identification of the essential role of stromal-derived factor 1 (SDF-1, including SDF-1α, SDF-1β isoforms) in the formation of metastasis in ECs and, at the same time, allowed us to conclude that kisspeptin-10 therapy is an effective inhibitor of metastasis, unfortunately only under the conditions of an in vitro experiment [28,29,30].

Much attention has also been paid to studying the expression of microRNAs (miRNAs) as markers of metastatic risk in EC. MicroRNAs (miRNAs) are small, non-coding RNA fragments that physiologically regulate gene expression under physiological conditions and also in pathologies such as oncogenesis and metastatic processes [31]. 

Abnormal, down-regulated miRNA expression has been found, among others, in the epithelial-to-mesenchymal transition (EMT) phenomenon, in which a miRNA (miR 199a/b 5p) regulates the expression of the key EMT gene FAM83B. Thus far, more than 150 miRNAs whose expression was downregulated in ECs and metastatic ECs (compared to normal endometrium) were found in databases. Under physiological conditions, miR 199a/b 5p inhibits EMT by regulating E cadherin, N cadherin, Snail, α smooth muscle actin, and vimentin expressions [32]. 

Another example of the effect of miRNAs on metastasis in ECs is miR-29a-3p. Low expression of this miRNA is statistically significantly associated with a high risk of disease progression. The molecular mechanism for this phenomenon involves miR-29a-3p’s failure to inhibit the VEGFA/CDC42/PAK1 proliferative pathway [33]. 

Research is underway to develop a miRNA profile to predict the risk of EC metastasis to lymph nodes with high precision (80–90%) [34].

An animal (mouse) model (orthotopic endometrial cancer mouse model) has also been created to study the anatomical and molecular mechanisms of distant metastasis formation in EC. 

Another factor connected with metastasis in EC includes overexpression of the transcription factor RUNX1, which was found to be associated with the formation of EC lung metastasis and nodal metastasis [35] (Table 1).

## 4. Imaging in Metastasis

Transvaginal ultrasound is the basic method for evaluating endometrial pathology, including cancer, with high diagnostic sensitivity, but this method of imaging may also be used as a rough prognostic factor [36], of course, verified later by histological examination.

Infiltration of the myometrium is the primary prognostic factor, as more than 70% of 5-year survival rates are recorded in cases of superficial infiltration (grade IA). This percentage decreases to 34% in cases of deep infiltration (stage IB), which promotes metastasis spread by both lymphatic and blood routes. In stages IA and IB, the incidence of metastasis to pelvic and/or paraortic lymph nodes is believed to be 10–12% and 30–60%, respectively. An important prognostic factor associated with an increased risk of lymph node metastasis is the histological differentiation of cancer (grading). In low-grade ECs, tumor infiltration of the uterine wall is found in 20% of cases, while in high-grade ECs, it is found in more than 50% [37].

Multicenter prospective studies defining EC-specific sonomorphologic and Doppler features have shown that low-risk (G1, FIGO IA) endometrial cancers are most often characterized by homogeneous echogenicity with no or minimal vasculature: 1–2 points according to IETA, while tumors with differentiated echogenicity with abundant perfusion of 3–4 points according to IETA criteria and numerous vessels penetrating multifocally from the myometrium into the uterine surface are more common in high-grade carcinomas (G2, G3,) with deep infiltration of the myometrium and/or cervical stroma involvement. Thus, these parameters can be further used in predicting the risk of lymph node metastasis and recurrence. It has been shown that there is a correlation between the type of vasculature and histopathological type, stage (FIGO), and risk of recurrence. The evaluation of quantitative parameters (RI, PSV, severity of vascularization) and qualitative parameters (type of echogenicity) due to their high utility is a valuable part of preoperative evaluation. Low opacities (RI < 0.4) and increased flow velocity (PSV) show a correlation with deep infiltration (uMI > 50%), high grade (G2–3), cervical infiltration, and LVSI involvement, indicating a high risk of lymph node metastasis and recurrence [38,39,40].

Therefore, the preoperative tumor grading, the depth of the uterine myometrium infiltration examined by transvaginal ultrasound, and, as reported by some, the serum level of CA 125 can be considered predictors of lymph node metastasis, as they have been shown to have a sensitivity and specificity of 94 and 57%, respectively [41].

Additionally, tumor size >2.5 cm, uMI > 50%, and RI < 0.4 correlate with the presence of lymph node metastasis [42]. This confirms the potential of ultrasound measurement of tumor size to predict unfavorable outcomes before surgery. In addition, tumor dimensions on preoperative ultrasound have been shown to be an independent predictor of recurrence or disease progression, even among cases classified as low-risk tumors [43].

Numerous studies have shown that such a two-step strategy combining preoperative histopathologic evaluation with ultrasound evaluation of myometrial and cervical stroma invasion helps identify cases at high and low risk of lymph node metastasis with a sensitivity and specificity of 83% and 71%, respectively [44,45].

An additional ultrasound parameter for preoperative assessment of the risk of lymph node metastasis is the measurement of the thickness of the unchanged (free) myometrium from the edge of the tumor infiltrate to the serosal membrane, known as uTFD (tumor-free distance) [46,47]. 

Both uMI and uTFD, either alone or in combination, are valuable in obtaining additional preoperative information about lymph node status. According to the data, the measurement of uTFD is more accurate than uMI in assessing locoregional cancer invasion. Therefore, uTFD may be recommended as an additional parameter, especially when uMI measurement alone is difficult, such as with the coexistence of myomas and/or adenomyosis [48,49,50]. 

Tumor size combined with p53 status on ultrasound allows preoperative identification of women at risk of recurrence or progression. As highlighted above, tumor size <2 cm as assessed by ultrasound combined with the absence of abnormal protein 53 (p53 abn) of the ProMisE subtype allows the identification of a large group (approximately 50%) of women with a very low risk of recurrence or disease progression, in whom sentinel node biopsy or adjuvant treatment is unnecessary.

Tumor size has also been correlated with lymph node involvement, as reported by Boyraz et al., who suggested that a tumor size greater than 2 cm could serve as an independent predictor of lymph node metastasis in low-risk EC patients. Mariani et al. observed no lymph node metastases among patients with a primary tumor diameter of ≤2 cm. Vargas et al. found that the rate of lymph node involvement increased from 1.3% in grade 1 and 3.8% in grade 2 tumors ≤2 cm to 12.7% in grade 1 and 23% in grade 2 tumors ≥5 cm, irrespective of myometrial invasion. Additionally, Cox-Bauer et al. reported that a cutoff of 5 cm was significantly more predictive of nodal involvement than a tumor diameter of 2 cm. Considering the above findings, the importance of tumor size as a risk factor for metastasis in EC patients is evident. Larger tumor sizes are consistently associated with adverse histological features and an increased likelihood of lymph node involvement. These observations underline the potential role of tumor size as a crucial prognostic factor in determining the metastatic risk in EC patients.

The combination of demographic characteristics (age, waist circumference, BMI), ultrasound findings, and ProMisE subtype shows a higher preoperative ability to predict a recurrence or progression of the disease than the ESMO risk classification, confirming its use in preoperative risk stratification in women with endometrial cancer. This may permanently enter into the range of parameters for choice of treatment and determining prognosis in EC [51] (Table 2).

## 5. Hysteroscopy and Laparoscopy

The diagnosis of endometrial cancer is made on the basis of histopathological examination of material taken from the uterine cavity. The predominant method of obtaining such material is uterine curettage, and, in selected cases, hysteroscopy is performed [52,53]. Endometrial biopsy performed under visual guidance in hysteroscopy has become the gold standard for precise diagnosis and sometimes treatment in patients wishing to preserve the uterus due to reproduction reasons. Its sensitivity in detecting endometrial cancer ranges from 67 to 86.4%, and its specificity is 99.2% [54]. A standardized objective tool for the systematic assessment of the endometrium during hysteroscopy, known as the HYCA (hysteroscopic cancer) grading system, was introduced, significantly increasing the sensitivity and specificity of this method [55].

In addition to determining the biopsy site, hysteroscopic imaging allows for the differentiation of cervical mucosal infiltration from a protrusion of endometrial cancer into the canal, which is significant in the management of therapy [56]. 

For many years, there has been intense discussion on the impact of diagnostic and/or therapeutic methods on the course and prognosis of EC. The EC spreads locally, via the lymphatic and circulatory routes, and intraperitoneally. Intraperitoneal spread occurs through the Fallopian tubes, the lymphatic system, and intra-abdominal tumor foci. Hysteroscopy and laparoscopy are particularly interesting as diagnostic and therapeutic methods, during which medium and positive pressure are used.

Since the introduction of hysteroscopy to diagnose endometrial cancer, concerns have been raised about the possibility of the associated spread of tumor cells beyond the primary focus, i.e., into the abdominal cavity and vagina. The primary route of cancer cell spread from the uterus is through the Fallopian tubes. Another hypothetical possibility, which remains unconfirmed practically, is vaginal infiltration with the medium during hysteroscopy, especially during surgical hysteroscopy.

Tests for the presence of endometrial cancer cells in the peritoneal cavity after hysteroscopy have been conducted for many years, and the results have not been consistent. 

To date, it has not been conclusively established that positive cytology from the peritoneal cavity is due to the diagnostic procedure performed, especially hysteroscopy, and not to a primary intraperitoneal dissemination [57,58].

However, a meta-analysis involving nearly 3000 women diagnosed with endometrial cancer found that hysteroscopy can increase the risk of metastasis of tumor cells to the peritoneal cavity, especially in advanced stages of cancer [57].

An increased incidence of positive cytology from the peritoneal cavity after diagnostic hysteroscopy has also been found in type II endometrial cancer [59]. 

The passage of cancer cells outside the uterine cavity during hysteroscopy may be influenced by the pressure used during this procedure. Theoretically, the pressure used during hysteroscopy can cause tumor cells to move not only through the Fallopian tubes but also into the uterine muscle and the lymphatic vessels. Several published studies, including a meta-analysis of nine clinical trials, have shown that pressures above 100 mm Hg increase the risk of tumor spread outside of the uterus [60]. 

According to the results of published studies, the data on the type of medium used during hysteroscopy and its relationship to tumor cell migration into the abdominal cavity are inconclusive. However, there are reports, both showing and denying, that the use of a liquid medium is associated with a higher risk of tumor cell outflow through the Fallopian tubes when compared to carbon dioxide [61].

A prospective study found that with an intrauterine pressure of less than 40 mm Hg, there is no leakage of fluid medium through the Fallopian tubes. Moreover, it was shown that the leakage of fluid medium through the Fallopian tubes is not connected with the spread of cancer cells in the peritoneal cavity [62]. 

In contrast, a randomized trial found that diagnostic hysteroscopy performed with a fluid medium at pressures below 70 mm Hg does not cause intraperitoneal spread of endometrial cancer and does not increase the risk of cancer recurrence during a 5-year follow-up [63].

The effect of the duration of hysteroscopy in patients with endometrial cancer on the course of the disease was examined in a retrospective multicenter study, and it showed no negative effect of prolonged hysteroscopy on positive peritoneal cytology and the incidence of cancer recurrence [64]. 

The formation of intraperitoneal metastases by cells that have entered the peritoneal cavity from the primary focus in the uterine cavity depends on the metastatic potential of these cells and local conditions. It has been found that EC cells that spread beyond the uterine cavity during hysteroscopy are 90% alive and capable of attaching to the substrate, but this was observed using an in vitro model [65]. 

The experts’ opinion claimed that the intraperitoneal presence of tumor cells is associated with an unfavorable prognosis only in cases where tumor foci are already present outside the uterus, but most also believe that the recurrence of endometrial cancer within one year after surgical treatment is due to the spread of tumor cells during the hysteroscopy performed [66,67].

A recently published systematic review and meta-analysis, which aimed to determine the oncological safety of hysteroscopy performed in early endometrial cancer compared to other diagnostic methods, included a group of 3980 patients with early EC. Hysteroscopy was performed on 1357 patients as a part of the diagnostic process. Based on data from two randomized trials and six cohort studies, it was determined that there were no differences in overall survival, disease-free survival, and disease-specific survival in the group of women who had undergone hysteroscopy compared to the group who had not. This indicates the complete oncological safety of the hysteroscopic procedure in early endometrial cancer [68].

Similar conclusions were drawn from a retrospective multicenter study, which compared two groups of patients with endometrial cancer that did not differ by age, BMI, tumor stage, histological type, peritoneal cytology result, depth of myometrial invasion, infiltration of lymphovascular spaces, and lymph node status. There were no statistically significant differences in overall survival and recurrence-free time between patients whose diagnosis of endometrial cancer was made based on histopathological examination of material from hysteroscopic biopsy and classical uterine curettage [69].

A retrospective comparison of hysteroscopic morcellation with other methods of endometrial biopsy; that is, hysteroscopy without morcellation, aspiration biopsy, and fractionated uterine curettage in women with endometrial cancer stage I to III, low to high (1–3) grade, endometrioid and non-endometrioid types, did not show that hysteroscopy with intrauterine morcellation was associated with increased spread of tumor cells, a higher incidence of infiltration of lymphovascular space or led to an increase in tumor grade in postoperative sample [70].

Laparoscopic removal of the uterus is an established surgical treatment for endometrial cancer, especially in the early stages. Randomized and observational studies have confirmed the lack of differences in 5-year survival compared to traditional hysterectomy. The vast majority of reports indicate significant advantages of this surgical method, especially in obese women. The spread of tumor cells during a hysterectomy performed by minimally invasive techniques may occur due to the use of a uterine manipulator, extraction of the uterus via the vaginal route, positive abdominal pressure, or the properties of carbon dioxide itself.

A retrospective study found no effect of laparoscopic surgery on survival, regardless of the absence or presence of cancer cells in peritoneal washings [71]. Recently (2022), a systematic review and meta-analysis were published on the effect of uterine manipulator use during minimally invasive surgery on oncological outcomes of EC. Thirteen retrospective studies, three prospective studies, and two randomized trials were analyzed. There were no differences in the incidence of infiltration of lymphovascular spaces in the manipulator-operated and traditional surgery or between the laparoscopic-operated group with or without the manipulator. There were no significant differences in the incidence of recurrence and positive cytology in women operated on with the manipulator compared to those without it. There were significant differences in positive cytology between those operated on traditionally and laparoscopically [72].

Metastasis at the trocar insertion site in early advanced endometrial cancer is rare, occurring at a rate of 0.18–0.33%, and is comparable to wound metastasis after conventional surgery. It is assumed that gas efflux around the trocar, improper surgical technique, the biological potential of metastatic cells, a reduced immune response at the site of abdominal incisions, the effect of carbon dioxide, and increased pressure on the peritoneal microenvironment may be responsible for their formation. Simple ways to reduce the risk of metastasis at trocar sites are described. These include avoiding excessive tissue traumatization, inserting only the necessary number of trocars, using heated and humidified carbon dioxide, flushing trocars with 5% iodinated povidone before guiding them into the abdominal cavity, fixing the trocars so that they do not dislodge, removing fluid and gas from the abdominal cavity before removing the trocars from their sites, removing surgical preparations in bags, excising the tumor with an adequate margin, and closing the peritoneum if the length of the incision exceeds 10–12 mm [73].

It can be summarized that another consequence of manipulator use, positive intra-abdominal pressure, and extraction of the uterus via the vaginal route, characteristic of laparoscopic hysterectomy, should be an increase in the incidence of implantation of cancer cells within the vagina and more frequent recurrence of endometrial cancer in the vaginal stump than after traditional hysterectomy. There are only a few reports on the implantation of endometrial cancer cells in the vagina after laparoscopic hysterectomy. Laparoscopic hysterectomy has also not been confirmed to increase the risk of endometrial cancer recurrence in the vagina stump compared to traditional hysterectomy [74] (Table 3.).

## 6. Discussion

Our exploration of hormonal and molecular aspects in endometrial cancer (EC) metastasis sheds light on critical factors influencing the progression of this disease. The multifaceted nature of metastasis, involving disturbances in cell adhesion to the extracellular matrix (ECM) and cytoskeletal abnormalities, underscores the complexity of this process, enabling cancer cells to migrate and invade secondary sites. The involvement of estrogens in actin and cell membrane remodeling through estrogen receptors (ER) and extragenomic interactions has significant implications. Molecular-level changes, such as moesin phosphorylation, contribute to enhanced migration and implantation capabilities of endometrial cells. Notably, the differential effects of tamoxifen and raloxifene on moesin highlight the clinical relevance of selective estrogen receptor modulators (SERMs) in modulating cytoskeletal dynamics. Studies further demonstrate estradiol’s association with EC proliferation, migration, and invasion through the activation of the IL-6 pathway. Prolactin emerges as a noteworthy player in EC metastasis, influencing chemotherapy sensitivity through complex endocrine, paracrine, and autocrine mechanisms. Both receptor and non-receptor modes of action contribute to anti-apoptotic effects, affecting the expression of Stat5a/b, Bcl-2, and Bax proteins. Prolactin’s stimulation of proliferation and its interaction with various ligands create a microenvironment conducive to proliferative processes. Additionally, prolactin directly stimulates vascular endothelial cell proliferation and indirectly influences proangiogenic factors. Pro-inflammatory adipocytokines, including leptin, visfatin, and resistin, play a role in the progression of endometrial cancer. Molecular insights reveal the significance of stromal-derived factor 1 (SDF-1) in metastasis formation, with experimental evidence supporting the inhibitory potential of kisspeptin-10. MicroRNAs (miRNAs) emerge as potential markers for metastatic risk in EC. Abnormal miRNA expression, especially in the context of the epithelial-to-mesenchymal transition (EMT), highlights their role in regulating key genes associated with cancer progression. The ongoing effort to develop a miRNA profile for predicting EC metastasis risk to lymph nodes underscores their potential clinical relevance. The creation of an orthotopic endometrial cancer mouse model provides valuable insights into the anatomical and molecular mechanisms underlying distant metastasis formation in EC. In conclusion, our findings unravel the intricate molecular landscape that influences endometrial cancer metastasis. Discussing these observations is paramount for understanding their implications, particularly in the context of well-established diagnostic and management modalities, such as ultrasound and hysteroscopy, as outlined in European and American guidelines. These insights contribute to a comprehensive understanding of EC progression, paving the way for informed clinical decisions and potential advancements in therapeutic strategies.

Myometrial invasion (DMI) plays a pivotal role in the staging and treatment decisions for endometrial cancer (EC) patients, contributing significantly to risk assessment and treatment planning. The integration of myometrial invasion status into the risk stratification model proposed by the ESTRO/ESGO/ESP guidelines, along with TCGA molecular groups, offers a comprehensive approach to prognostication in EC patients. The ESTRO/ESGO/ESP guidelines recommend considering myometrial invasion, along with other clinicopathologic factors, for risk stratification in EC patients. However, recent studies have emphasized the importance of integrating molecular factors, specifically the TCGA molecular groups, into the risk assessment process. These groups, including a subgroup characterized by mismatch repair gene deficiency, contribute valuable molecular insights that can enhance the accuracy of prognosis in conjunction with classic prognostic factors [18,19,51,75,76,77,78,79]. It is crucial to delve into the relationship between TCGA groups and myometrial invasion, given its prognostic significance and potential association with the risk of metastasis. While myometrial invasion is a well-established prognostic factor, understanding its interplay with molecular factors can further refine risk assessment and guide treatment decisions. Raffone et al.’s study provides evidence that myometrial invasion does not independently impact overall survival (OS) in EC patients [80]. However, it appears to influence the risk of recurrence, demonstrating its relevance in the context of TCGA molecular groups. The findings indicate a need for further exploration and confirmation through additional studies that specifically assess the prognostic impact of myometrial invasion within each TCGA group. Moreover, the discussion should extend to include other histopathological factors considered in the guidelines, as their prognostic significance may vary across TCGA groups. For instance, the reproducibility and prognostic value of lymphovascular space invasion (LVSI) have been highlighted in the literature, reinforcing its role as an independent prognostic factor [15,46]. Additionally, emerging factors, like the microcystic, elongated, and fragmented (MELF) pattern of invasion and tumor budding, warrant further investigation for their potential integration into the risk stratification system. In conclusion, while myometrial invasion remains a critical factor in risk assessment for EC patients, the integration of TCGA molecular groups provides a more nuanced understanding of prognosis. The discussion should emphasize the intricate relationship between myometrial invasion and molecular factors, paving the way for a comprehensive and tailored risk stratification model that optimally guides treatment decisions and improves patient outcomes. Further studies are warranted to validate these findings and explore the prognostic impact of myometrial invasion within each TCGA group, aligning with the evolving landscape of molecular-based risk assessment in endometrial cancer.

## 7. Conclusions

According to the presented data, it is reasonable to introduce ultrasound and hysteroscopy into daily practice as first-line methods in the diagnostic process of EC. Also important is the use of molecular profiling of EC to choose adequate surgery, sometimes adjuvant therapy, and to have a more precise prognosis for these patients.

## Figures and Tables

**Table 1 cancers-16-00179-t001:** Concise summary of the various aspects related to metastasis in endometrial cancer.

Aspect	Summary
Metastasis Elements	Disturbances in cell-ECM adhesion, cytoskeleton disruptions, and invasive properties are crucial in cancer metastasis.
Influence of Estrogen	Estrogen affects actin and cell membranes, increasing migration and implantation capabilities in endometrial cells.
Role of Prolactin	Prolactin has complex actions, including anti-apoptotic effects, proliferation stimulation, and angiogenic influence in EC.
Involvement of Adipocytokines	Adipocytokines like leptin, visfatin, and resistin contribute to the progression and spread of endometrial cancer cells.
SDF-1 and Kisspeptin-10	SDF-1 plays a crucial role in EC metastasis, and kisspeptin-10 therapy shows promise as a metastasis inhibitor (in vitro).
MicroRNAs (miRNAs)	Abnormal miRNA expression, such as miR-199a/b 5p and miR-29a-3p, is linked to EC metastasis, and profiles are under development.
RUNX1 and Metastasis	Overexpression of the transcription factor RUNX1 is associated with EC lung and nodal metastasis.

**Table 2 cancers-16-00179-t002:** Key points related to the role of transvaginal ultrasound (TVUS) in evaluating endometrial cancer and its prognostic factors.

Aspect	Summary
Role of Transvaginal Ultrasound (TVUS)	TVUS is fundamental in evaluating endometrial pathology, serving as a preliminary prognostic factor later confirmed by histological examination.
Prognostic Factors in Endometrial Cancer	Myometrial infiltration depth is a primary prognostic factor; deep infiltration (stage IB) is associated with a higher risk of metastasis. Histological differentiation (grading) is also a prognostic factor.
Sonomorphologic and Doppler Features	Low-risk endometrial cancers exhibit homogeneous echogenicity with minimal vasculature, while high-grade tumors show differentiated echogenicity and abundant perfusion. These parameters can predict lymph node metastasis and recurrence.
Ultrasound Parameters for Prediction	Preoperative grading, myometrial infiltration depth (uMI), and serum CA 125 levels can predict lymph node metastasis. Tumor size > 2.5 cm, uMI > 50%, RI < 0.4, and uTFD correlate with metastasis and unfavorable outcomes.
Two-Step Strategy	Combining preoperative histopathologic evaluation with ultrasound assessment of myometrial and cervical stroma invasion aids in identifying high and low-risk cases of lymph node metastasis.
Tumor-Free Distance (uTFD)	uTFD measurement is valuable for assessing locoregional cancer invasion and may be recommended alongside uMI, especially when myomas or adenomyosis coexist.
Ultrasound for Risk Identification	Tumor size and p53 status assessment on ultrasound help identify women at risk of recurrence or progression, aiding in treatment decisions and prognosis in endometrial cancer.
Combined Demographic and Ultrasound Factors	Combining demographic factors, ultrasound findings, and ProMisE subtype improves preoperative risk stratification, outperforming traditional risk classifications in endometrial cancer.

**Table 3 cancers-16-00179-t003:** Summary of the key points related to endometrial cancer diagnosis and the impact of diagnostic and therapeutic methods on the disease.

Aspect	Summary
Diagnosis of Endometrial Cancer	Diagnosis primarily involves histopathological examination of uterine cavity material. Hysteroscopy-guided endometrial biopsy is the gold standard for precise diagnosis (sensitivity: 67–86.4%, specificity: 99.2%).
HYCA Grading System	The HYCA grading system enhances the sensitivity and specificity of hysteroscopy for endometrial cancer diagnosis.
Influence of Diagnostic Methods on Spread	Concerns exist regarding potential tumor spread during diagnostic methods like hysteroscopy, especially in advanced cancer stages.
Pressure During Hysteroscopy	Pressure exceeding 100 mm Hg during hysteroscopy may increase the risk of tumor spread beyond the uterus.
Medium Used in Hysteroscopy	The impact of the type of medium (liquid vs. carbon dioxide) on tumor cell migration during hysteroscopy remains inconclusive.
Peritoneal Cavity Cytology	Studies on peritoneal cytology after hysteroscopy have yielded inconsistent results regarding its link to diagnostic procedures.
Effect on Prognosis	Positive cytology after hysteroscopy may be associated with an unfavorable prognosis, especially when tumor foci are already present outside the uterus.
Oncological Safety of Hysteroscopy	Recent systematic reviews and meta-analyses suggest the oncological safety of hysteroscopy in early endometrial cancer diagnosis.
Laparoscopic Hysterectomy	Laparoscopic hysterectomy is an established treatment for endometrial cancer; spread risk is associated with various factors such as uterine manipulator use and pressure.
Metastasis at Trocar Insertion Site	Rare metastasis occurs at trocar insertion sites during laparoscopic procedures; measures to reduce risk include minimizing tissue trauma and using proper techniques.
Implantation in the Vaginal Stump	Laparoscopic hysterectomy does not significantly increase the risk of implantation in the vaginal stump or endometrial cancer recurrence compared to traditional hysterectomy.

## Data Availability

No new data were created or analyzed in this study. Data sharing is not applicable to this article.

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
