# Peer review of "Metastases and Recurrence Risk Factors in Endometrial Cancer—The Role of Selected Molecular Changes, Hormonal Factors, Diagnostic Methods and Surgery Procedures"

_cancers, 2023, doi:10.3390/cancers16010179_

Round 1
Reviewer 1 Report
Comments and Suggestions for Authors
This review tries to cover many aspects of endometrial cancer starting from diagnosis, imaging, hormonal factors and surgery procedures and risk factors for metastases. While topic is important, a number of good reviews have been published in recent months. Limiting to metastatic disease may do better justice on the topic. Following suggestions may be considered.
1. Lot of information has been provided under various subheadings but how a reader have to use this information in a given case is not clear. if this part could be elaborated with example of a real case in different settings it will improve the importance of the topic.
Comments on the Quality of English Language
Flow of thoughts is missing If authors can work little more to improve the spontaneity., this will improve quality of the review.
Author Response
Thank you very much for this suggestion. Indeed, recently there have been many publications on endometrial cancer, primarily associated with the introduction of molecular classification. However, none of them focus on the topic of metastasis in this cancer. This topic will probably remain relevant for a long time until we ultimately establish all the relationships between molecular diagnostics and various clinical situations.
In our clinical data, we do not have a case that would significantly enrich the content of this publication.
Reviewer 2 Report
Comments and Suggestions for Authors
The aim of this article was to describe various factors contributing to development of metastasis in endometrial carcinoma (EC), including the cancer's molecular profile, hormone activity (like estrogen and prolactin), and pro-inflammatory adipocytokines. It also investigates how altered microRNA expression affects gene regulation linked to cancer spread.
Unfortunately, it is not possible to accept the article in the present form due to major flaws:
· Introduction: after a description of the background, authors should explain the objective of their review and the reasons why they wrote this article.
· Authors should describe materials and methods they used to write the review. What time of review it is, how they conducted the literature search and how were selected the included studies.
· Authors should discuss their finding and the implications of their observations in a specific section, focusing in particular on the molecular aspects, since ultrasound and hysteroscopy are already widely accepted in EC diagnosis and management in European and American guidelines.
· Myometrial invasion is a crucial factor in determining the stage and treatment plan for endometrial cancer patients, and accurate assessment is essential for optimal patient care. The ESTRO/ESGO/ESP guidelines for the management of EC proposed a novel risk stratification model including molecular TCGA molecular groups (which include a group characterized by mismatch repair genes deficiency) to assess the prognosis of EC in association with classic, well-known, clinicopathologic prognostic factors of EC (such as myometrial invasion, histotype or lymph vascular space invasion). Some recent studies analyzed the relationship between TCGA groups and classic prognostic factors (grade, myometrial invasion, LVSI) and I believe the discussion should include the relation between TCGA groups and myometrial invasion, as it represents a prognostic factor and could also be related to the risk of metastasis (e.g. PMID: 34088515).
- In the ultrasound section, Authors may discuss about descripted preoperative ultrasound characteristics of EC, such as Karlsson’s Ratio (see PMID: 28812010) or tumor size (see PMID: 36212493) as possible factors for the risk of metastasis, since they were described as prognostic factors.
Comments on the Quality of English Language
English needs to be extensively revised by a native speaker
Author Response
Thank you for taking the time to review our manuscript. We appreciate your insightful comments and suggestions, and we are pleased to inform you that we have carefully addressed each of your points in the revised manuscript. We added to the background section some more explanation i.d. “In light of the observed trend in recent years related to the incidence, and especially the mortality, of endometrial cancer both in Poland and worldwide, we have decided to assess factors that may be associated with a higher likelihood of metastasis occurrence, thereby influencing the increase in mortality among women with endometrial cancer”.
Regarding the materials and methods used for the review, we have incorporated a dedicated section that outlines the type of review, details on the literature search methodology, and the criteria for selecting included studies. This addition enhances the transparency of our approach and provides clarity on the methodology employed in writing the review.
In response to your suggestion about discussing the findings and implications, particularly focusing on molecular aspects, we have included a discussion that delves into the molecular dimensions of our observations. This addition contributes to a more comprehensive discussion, aligning with the guidelines and emphasizing the molecular aspects in the context of widely accepted diagnostic and management practices in endometrial cancer.
Regarding your valuable point about the ESTRO/ESGO/ESP guidelines and the relation between TCGA groups and myometrial invasion, we have expanded our discussion to explicitly address this relationship. We've explored how myometrial invasion, as a crucial prognostic factor, intersects with the molecular TCGA groups, offering a more nuanced understanding of its implications and potential relevance to the risk of metastasis. We have incorporated relevant references, to enrich the discussion.
Concerning the ultrasound section, we acknowledge your suggestion to discuss Karlsson's Ratio and tumor size as potential prognostic factors for the risk of metastasis. However, after a thorough search, we could not find information on Karlsson's Ratio in the context of endometrial cancer and metastasis.
Reviewer 3 Report
Comments and Suggestions for Authors
I congratulate the authors for their work, which is very complete and detailed.
The treatment of the topic is absolutely impeccable.
The exposition is clear, the data is complete and updated.
Some small typos need to be reviewed (excessive spaces, missing punctuation, etc.) and on page 7, the table has a typo on the first line on the left.
Author Response
We did all necessary corrections according reviewer suggestions - excessive spaces, missing punctuation, etc.
Reviewer 4 Report
Comments and Suggestions for Authors
Markowska et al review several aspect related with metastasis development and diagnosis in endometrial cancer. Although at present the molecular classification of EC is well established, some comments about the differential metastatic potential of histological subtypes should be presented. In addition, the author should comment in more details studies analyzing the participation of specific genes and pathways in the process of EC metastasis: are genes and pathways involved in EC initiation also involved in metastatic progression (PTEN, CTNNB1, TP53 and other genes; PIK3, WNT and other pathways? Are the same genes and pathways involved in the progression of different histological types (ie. serous vs endometrioid carcinomas).
Author Response
So far, there have not been many publications concerning the association between the presence of metastases and molecular changes, apart from assessing this relationship at various clinical stages and, consequently, its connection with the presence of metastases within lymph nodes. In the publication submitted for review, we have described available scientific data regarding molecular changes associated with metastasis (SDF-1, microRNAs, VEGFA/CDC42/PAK1, RUNX1).
However, we appreciate the valuable comments regarding the possibility of the association of metastasis in endometrial cancer with other molecular pathways.
PTEN – phosphatase and tensin homolog
The tumor suppressor gene (PTEN) was identified in 1997; it is located on chromosome 10 (10q23). PTEN inhibits cell proliferation and differentiation, and it participates in the insulin signaling pathway. The protein encoded by this gene is a 55 kDa protein consisting of 403 amino acids, exhibiting tyrosine phosphatase activity. PTEN protein negatively regulates the phosphatidylinositol 3-kinase (PI3K) signaling pathway. The effector downstream of PI3K is the Akt protein, which is a serine-threonine kinase. Therefore, the PTEN protein can function through the Akt signaling pathway [Scully, Erkanii].
In the publication by Stavropoulos et al., an analysis was conducted on the association of PTEN expression with clinical data and the co-occurrence of changes in p53 expression. The intensity of PTEN expression was not significantly associated with the mean age of patients (p=0.387), histological type of the tumor (p=0.630), depth of myometrial invasion (p=0.124), clinical stage (p=0.621), vascular invasion (p=0.442), presence of necrosis in the tumor (p=1.000), or fallopian tube invasion (p=0.524). Furthermore, the results suggested that there was no significant correlation between the intensity of PTEN staining and histological grade (p=0.071). Strong positive PTEN expression was observed in 4 cases (20.0%) of histological grade G1, 21 cases (42.9%) of grade G2, and 5 cases (16.7%) of histological grade G3. Correspondingly, the frequencies for moderate expression were as follows: 9 (45.0%), 17 (34.7%), and 14 (46.7%) [Stavropoulos]. Data regarding the association of PTEN with metastasis and prognosis are inconclusive, but the majority indicates a link between loss of PTEN expression and favorable prognosis [Chaopeng].
Scully MM, Palacios-Helgeson LK, Wah LS and Jackson TA: Rapid estrogen signaling negatively regulates PTEN activity through phosphorylation in endometrial cancer cells. Horm Cancer. 5:218–231. 2014. View Article : Google Scholar : PubMed/NCB
Erkanli S, Kayaselcuk F, Kuscu E, Bagis T, Bolat F, Haberal A and Demirhan B: Expression of survivin, PTEN and p27 in normal, hyperplastic, and carcinomatous endometrium. Int J Gynecol Cancer. 16:1412–1418. 2006. View Article : Google Scholar : PubMed/NCBI
Stavropoulos A, Varras M, Vasilakaki T, Varra VK, Tsavari A, Varra FN, Nonni A, Kavantzas N and Lazaris AC: Expression of p53 and PTEN in human primary endometrial carcinomas: Clinicopathological and immunohistochemical analysis and study of their concomitant expression. Oncol Lett 17: 4575-4589, 2019
Chaopeng Ye, Yuanqiao Ma, Qi Zhang et al. Prognostic Analysis of PTEN Mutation in Endometrial Carcinoma, 24 January 2022, PREPRINT (Version 1) available at Research Square [https://doi.org/10.21203/rs.3.rs-1252352/v1).
CTNNB1
Hyperactivation of the Wingless/int1 (Wnt)/β-catenin signaling pathway is associated with carcinogenesis, tumor progression, disease recurrence, and chemotherapy resistance in malignant gynecological tumors. Specifically, Wnt signaling promotes metastasis and therapy resistance in ovarian cancer, plays a crucial role in carcinogenesis and recurrence in endometrial cancer, and is involved in the carcinogenesis and metastasis associated with human papillomavirus (HPV) in cervical cancer.
Among the more common genetic mutations in endometrial cancer (EC) are alterations in the beta-1 catenin or β-catenin gene (CTNNB1), occurring in approximately 20-25% of tumors (McMellen). Catenins constitute a group of three cytoplasmic protein subtypes (α, β, and γ) that interact with cadherins. Both β-catenin and plakoglobin (γ-catenin) form a connection between the cytoplasmic domain of E-, N-, and P-cadherins and α-catenin, which binds to actin cytoskeletal filaments. Therefore, β-catenin plays a crucial role in cell adhesion, but in addition to its function as a linking protein, it also plays a significant role in the Wingless/int1 (Wnt) signal transduction pathway, regulating cell proliferation and differentiation [Shapiro, Bensal].
Retrospective analyses indicate that mutations in CTNNB1 are associated with poorer recurrence-free survival and may be linked to a higher incidence of distant metastases. Based on the frequency of tumors with CTNNB1 mutation in the NSMP group (26–52%) with independent prognostic value, some authors suggest that tumors with CTNNB1 mutation should be considered as the fifth molecular subgroup [De Leo, Santoro].
McMellen A., Woodruff E. R., Corr B. R., Bitler B. G., Moroney M. R. Wnt signaling in gynecologic malignancies. International Journal of Molecular Sciences . 2020;21(12):1–21. doi: 10.3390/ijms21124272 ).
(Shapiro L., Weis W. I. Structure and biochemistry of cadherins and catenins. Cold Spring Harbor Perspectives in Biology . 2009;1(3) doi: 10.1101/CSHPERSPECT.A003053.
Bansal N., Yendluri V., Wenham R. M. The molecular biology of endometrial cancers and the implications for pathogenesis, classification, and targeted therapies. Cancer Control . 2009;16(1):8–13. doi: 10.1177/107327480901600102).
De Leo A., de Biase D., Lenzi J., et al. ARID1A and CTNNB1/β-catenin molecular status affects the clinicopathologic features and prognosis of endometrial carcinoma: implications for an improved surrogate molecular classification. Cancers. 2021;13(5):p. 950. doi: 10.3390/CANCERS13050950. [PMC free article] [PubMed] [CrossRef] [Google Scholar]
Santoro A., Angelico G., Travaglino A., et al. New pathological and clinical insights in endometrial cancer in view of the updated ESGO/ESTRO/ESP guidelines. Cancers . 2021;13(11):p. 2623. doi: 10.3390/CANCERS13112623. [PMC free article] [PubMed] [CrossRef] [Google Scholar]
Round 2
Reviewer 2 Report
Comments and Suggestions for Authors
I kindly thank Authors for answering to all my comments. I believe the manuscript has improved with revisions and is now suitable to be accepted.
Reviewer 4 Report
Comments and Suggestions for Authors
No additional comments.